cellular biology

RUFY, mitophagy, Skp, alveolar macrophages, LPS

**Authors for correspondence:**
Evelina Gatti
e-mail: gatti@ciml.univ-mrs.fr
Philippe Pierre
e-mail: pierre@ciml.univ-mrs.fr

†Equal contribution.

# RUFY4 exists as two translationally regulated isoforms, that localize to the mitochondrion in activated macrophages

Jan Valečka[1], Voahirana Camosseto[1], David G. McEwan[4],
Seigo Terawaki[5], Zhuangzhuang Liu[6], Eva Strock[1],
Catarina R. Almeida[2], Bing Su[3], Ivan Dikic[7],
Yinming Liang[6], Evelina Gatti[1,2,†] and
Philippe Pierre[1,2,3,†]

[1]Aix Marseille Université, CNRS, INSERM, CIML, 13288 Marseille cedex 9, France
[2]Institute for Research in Biomedicine (iBiMED) and Ilidio Pinho Foundation, Department of Medical Sciences, University of Aveiro, 3810-193 Aveiro, Portugal
[3]Shanghai Institute of Immunology, Department of Microbiology and Immunology, Shanghai Jiao Tong University School of Medicine, Shanghai 200025, People's Republic of China
[4]Tumour Cell Death Laboratory, Cancer Research UK Beatson Institute, Garscube Estate, Switchback Road, Glasgow G61 1BD, UK
[5]Department of Pathobiochemistry, Graduate School of Medicine, Osaka City University, Osaka, Japan
[6]School of Laboratory Medicine, Xinxiang Medical University, Xinxiang, People's Republic of China
[7]Institute of Biochemistry II, Goethe University School of Medicine, Theodor-Stern-Kai 7, 60590 Frankfurt (Main), Germany

DGM, 0000-0003-0353-6532; PP, 0000-0003-0863-8255

We report here that RUFY4, a newly characterized member of the 'RUN and FYVE domain-containing' family of proteins previously associated with autophagy enhancement, is highly expressed in alveolar macrophages (AM). We show that RUFY4 interacts with mitochondria upon stimulation by microbial-associated molecular patterns of AM and dendritic cells. RUFY4 interaction with mitochondria and other organelles is dependent on a previously uncharacterized OmpH domain located immediately upstream of its C-terminal FYVE domain. Further, we demonstrate that *rufy4* messenger RNA can be translated from an alternative translation initiation codon, giving rise to a N-terminally truncated form of the molecule lacking most of its RUN domain and with enhanced potential for its interaction

with mitochondria. Our observations point towards a role of RUFY4 in selective mitochondria clearance in activated phagocytes.

# 1. Introduction

The RUN and FYVE domain-containing proteins (RUFY) family encompass five conserved genes displaying tissue-specific expression [1]. The different RUFY proteins have been described to regulate endosomal trafficking, autophagy and cell migration. RUFY family protein dysfunction, can lead to severe pathologies, including cancer [1]. They share a common structural organization with an N-terminal RUN domain, several coiled-coil (CC) motifs and a PtdIns(3)P-interacting C-terminal FYVE domain. Distinct from other RUFY proteins, RUFY4 lacks the tandem histidine cluster and the SH3 binding domain that normally define consensus FYVE domains [2]. RUFY4 can nevertheless interact with phosphatidylinositol 3-phosphate (PtdIns(3)P)-enriched membranes [3] and upon overexpression, induce the degradation of the autophagy effector LC3/ATG8 together with the perinuclear clustering of late endosomal compartments and autophagosomes [4]. RUFY4 expression remains low in most cells and tissues with the exception of lungs and lymphoid organs. RUFY4 was found to be strongly induced *in vitro* in dendritic cells (DCs) differentiated from bone marrow progenitors in presence of granulocyte and macrophage colony-stimulating factor (GM-CSF) and interleukin 4 (IL-4) [4]. *In vivo*, its expression was confirmed in lung DCs isolated from asthmatic mice. RUFY4 seems, therefore, able to harness the classical macro-autophagy pathway (hereafter, autophagy) to facilitate autophagosome formation and increase autophagy flux [5,6]. By optimizing effector proteins activity and organelles distribution, RUFY4 expression facilitates the elimination of intracellular bacteria like *Brucella abortus*, and *Salmonella typhimurium* replication [4,7], suggesting that it has a role in the cell response to bacterial infection.

RUFY 4 is expressed in phagocytes that bear pattern recognition receptors capable of recognizing microbe-associated molecular patterns (MAMPs) [8]. Lipopolysaccharide (LPS) detection by Toll-like-receptor 4 (TLR4) triggers phagocyte activation through different signalling cascades resulting in secretion of pro-inflammatory cytokines, expression of surface co-stimulatory molecules. Activation also results in enhanced antigen processing and major histocompatibility class II restricted presentation of antigens derived both from intracellular or extracellular antigens and pathogens [9,10]. All these functions are accompanied by major remodelling of membrane trafficking and actin organization to favour both phagocytosis and migration to the lymphoid organs [11].

Herein, we investigate the regulation of RUFY4 expression in phagocytes upon MAMPs detection and show that RUFY4 is strongly expressed in alveolar macrophages (AM) *in vivo* [12]. AM and RAW 264.7 macrophages exposure to LPS and other innate stimuli induce the localization of RUFY4 to the mitochondrial network. Together with its reported association with Lamp1-positive organelles [4], a role for RUFY4 in late endosomes and potentially mitochondria regulation is further suggested by the identification by mass spectrometry (MS) of different interactors, such as Ras-related protein 34 (RAB34), Pleckstrin Homology and RUN Domain Containing M1 (PLEKHM1) or N-alpha-acetyltransferase 30 (NAA30). We further show that a putative 17 kDa protein Skp (also known as OmpH) domain present immediately upstream of the RUFY4 C-terminal FYVE domain is functional and promotes subcellular organelles aggregation and binding to mitochondria observed upon ectopic expression of RUFY4. We demonstrate that *rufy4* messenger RNA (mRNA) is also submitted to translational regulation through the use of an alternative translation initiation codon (AIC), that gives rise to a partially truncated isoform with an impaired RUN domain, that could potentially dimerizes with and regulates full-length RUFY4 function. All together our findings point at the existence of a previously unknown interaction of RUFY4 with the mitochondria, which is subject to a complex translational regulation during phagocyte activation by MAMPs.

# 2. Results

## 2.1. RUFY4 detection in macrophages is linked to microbial or type-I-IFN activation

Genomic databases interrogation suggests that the *rufy4* gene has only recently evolved as an independent member of the RUFY family expressed only in mammalian cells [1]. We identified an alternative transcript (v2) lacking most of exon 3, and like the main *rufy4* mRNA (v1), bears a

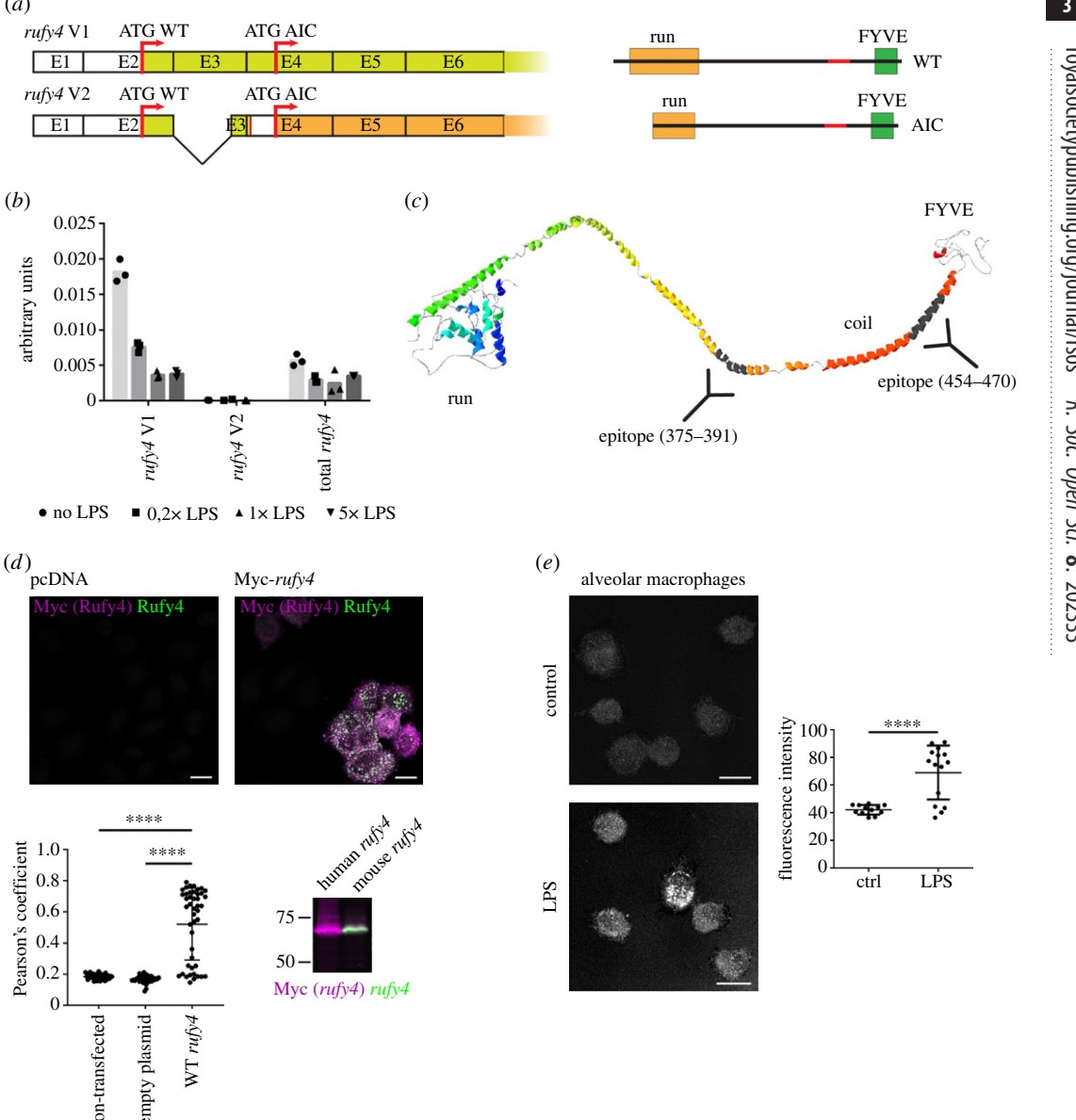

**Figure 1.** RUFY4 structure and expression in alveolar macrophages. (*a*) Scheme of mRNA and domain structure of two endogenously expressed transcripts variants of *rufy4*. (*b*) Mouse alveolar macrophages were activated by 0, 20, 100 or 500 ng ml$^{-1}$ of LPS for 8 h, and *rufy4* mRNA expression levels was analysed by qPCR, using transcript specific primers. (*c*) Phyre2 structure and domain prediction for RUFY4, antigenic epitopes used to raise an anti-*rufy4* antibody are indicated. (*d*) HeLa cells were transfected with empty plasmid (pcDNA) or Myc-tagged mouse *rufy4* and stained for Myc (magenta) and RUFY4 (green) prior visualization by immunofluorescence confocal microscopy (ICM) (scale bars 20 µm) or immunoblot. ICM co-localization (white) was quantified using Pearson's coefficient correlation in IMAGEJ and statistical relevance established by one-way ANOVA test, $^{****}p \leq 0.0001$, cells not expressing wild-type (WT) RUFY4, despite transfection, were included in the graph (lower points). HeLa cells were also transfected by human and mouse Myc-*rufy4* and analysed by immunoblot side to side after co-staining and immunofluorescence detection, co-detection of Myc (magenta) and RUFY4 (green) was only observed for the mouse RUFY4 (white) and not the human isoform (lilac). (*e*) Primary alveolar macrophages (AM) were activated with 100 ng ml$^{-1}$ of LPS for 8 h and stained for RUFY4 by immunofluorescence. Intensity of fluorescence was analysed in each cell. Scale bar 10 µm. Statistical relevance was analysed by Student's *t*-test, $^{****}p \leq 0.0001$.

predicted AIC [13] at the beginning of exon 4 (figure 1*a*). Interestingly, a transcript completely lacking the RUN domain and likely to give rise to a protein equivalent to the putative AIC form was also identified in human genomic databases (electronic supplementary material, figure S1A). Expression of mouse *rufy4* mRNA was previously shown to be GM-CSF and IL-4 dependent [4]. Given the dependency on GM-CSF

for AM development [14], we monitored *rufy4* transcription in freshly isolated mouse AM from bronchoalveolar lavage. Expression of *rufy4* mRNA was found to be extremely high in AM and decreased upon LPS activation, with only the full-length *rufy4* isoform (v1) detected by quantitative polymerase chain reaction (qPCR) (figure 1*b*).

The absence of an adapted reagent for RUFY4 detection, led us to raise a new polyclonal antibody (pAb) against the peptides 375–391 and 454–470 of mouse RUFY4, that presents a high probability of epitope accessibility based on the predicted RUFY4 structure (figure 1*c*), calculated using the Phyre2 web portal [15]. Affinity-purified antibodies efficiently detected RUFY4 as demonstrated by a strong staining overlap upon detection by immunofluorescence confocal microscopy (ICM) of a transfected myc-tagged form of the molecule with the novel anti-RUFY4 and anti-Myc tag antibodies (figure 1*d*). The specificity of the antibody for mouse RUFY4 was confirmed upon detection by immunoblot of over-expressed myc-tagged mouse RUFY4, but not of the equivalent human isoform (figure 1*d*). We were, however, unable to detect by standard immunoblots the physiological levels of RUFY4. As an alternative, we turned to ICM to monitor RUFY4 expression in mouse AM stimulated or not with LPS (figure 1*e*). Expression of RUFY4 could only be clearly detected in LPS-stimulated cells, although unstimulated AM expressed an already high amount of the *rufy4* mRNA (figure 1*b*). This difference suggests that independently of transcription, other regulatory mechanisms linked to MAMPs-dependent activation might be involved with RUFY4 expression and/or localization.

RUFY4 regulation was next monitored in RAW macrophages, in which RUFY4 detection was again increased upon LPS stimulation with a clear localization to subcellular organelles distributed throughout the cytosol (figure 2*a*). As an alternative to LPS treatment, AM were exposed to type-I interferon (IFN) either by adding directly recombinant IFN-α to the culture media, or by transfecting the cells with plasmid DNA (pcDNA), which activates the cGAS/STING pathway [16] and leads to type-I IFN release and indirect cell activation. With all chosen stimuli, RUFY4 was found to accumulate in a pattern suggesting again an association to subcellular organelles (figure 2*b*). These observations could be recapitulated in IL4/GM-CSF bone-marrow-derived DCs (figure 2*c*), further suggesting that RUFY4 activity is regulated by MAMPs sensing, in agreement with its proposed role for intracellular bacteria elimination [4,7]. Interestingly, the total fluorescent staining intensity detected in AM and DC was not significantly increased by activation (figure 2*b*,*c*), in contrast to the decrease in mRNA levels previously observed in these conditions (figure 1*b*) [4]. This suggests that enhanced detection of RUFY4 upon activation is preferentially because of changes in its subcellular localization that concentrate or reveal the epitope(s) recognized by the antibodies, rather than solely mRNA expression and/or translation enhancement.

## 2.2. Ectopically expressed RUFY4 interacts with PLEKHM1 and mitochondria-associated molecules

We next probed RUFY4 interactome by performing stable isotope labelling with amino acids in cell culture (SILAC), prior immunoprecipitation and comparative analysis by MS of RUFY4 interacting partners using HeLa cells stably expressing an mCherry-flag-tagged RUFY4 fusion protein. We first established that stable RUFY4 chimera expression augmented LC3-II autophagic flux at steady state and upon chloroquine treatment [17] (figure 3*a*). We also showed that the construct induced and co-localized partially with Lamp1-positive late endosome and lysosomes perinuclear clusters (figure 3*a*). Immunoprecipitation of mCherry from SILAC treated cells was performed prior to SDS–PAGE and trypsin digestion. SILAC ratios (heavy/light; H/L) of identified peptides were then established by MS. A normalized H/L ratio ≥ 1.5 was used as a cut-off to identify potential RUFY4-mCherry interaction partners (figure 3*b*; electronic supplementary material, table 1). Out of 50 identified proteins, six displayed high-H/L ratio ≥ 2.6 (figure 3*b*), including RAB34, PLEKHM1 and NAA30, all involved in late endosomal membranes regulation, positioning or fusion [18–21]. Gene ontology analysis using the G-Profiler program [22] confirmed with high confidence ($p_{adj} = 3.14 \times 10^{-6}$) that most putative RUFY4 interacting partners were associated with intracellular membranes including endosomes, endoplasmic reticulum (ER) and Golgi (figure 3*c*). The presence of HOPS subunit VSP39, RAB34, YKT6 and of PLEKHM1 in the list, could be easily linked to the late endosome/lysosome tethering capacity of RUFY4, given the reported implication of all these molecules in this process [23], preceding homotypic fusion or autophagolysosomes formation. Moreover, in addition to NAA30, which is also essential for mitochondrial integrity and function [24], eight of RUFY4 putative partners are directly associated with mitochondria (electronic supplementary material, table S1), suggesting



(*a*) RAW cells

control          LPS 8 h

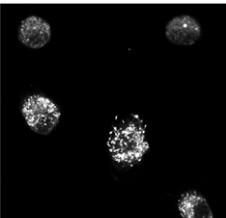

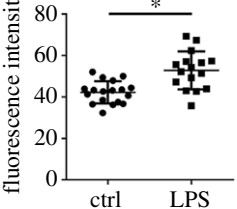

(*b*) alveolar macrophages

control          pcDNA

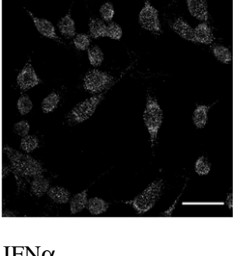

IFNα

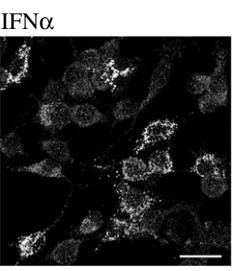

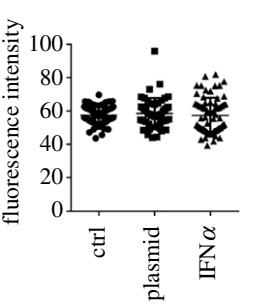

(*c*) IL-4 BMDCs

control          LPS 6 h

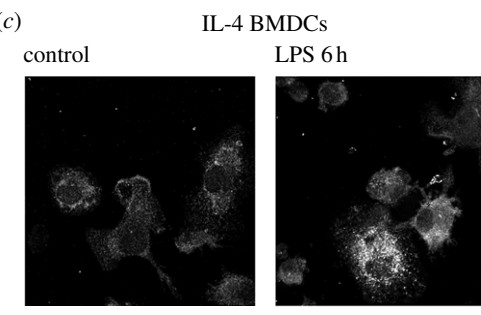

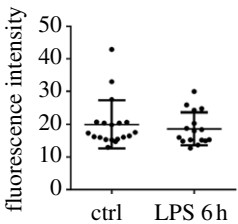

**Figure 2.** Rufy4 labelling increased upon activation. (*a*) RAW macrophages were activated by 100 ng ml$^{-1}$ of LPS for 8 h and RUFY4 was imaged by ICM. Fluorescence intensity was analysed in each cell. Scale bar 20 µm. Statistical analysis was performed using Student's *t*-test, $^{*}p \leq 0.05$. (*b*) Mouse alveolar macrophages were activated by transfection of empty pcDNA3.1 plasmid or by IFN-α and stained for RUFY4 prior to ICM. Scale bar 20 µm. Statistical analysis was performed using one-way ANOVA, $p > 0.05$. (*c*) Bone marrow-derived dendritic cells grown in IL-4 were activated by 100 ng ml$^{-1}$ LPS for 6 h and stained for RUFY4 prior to ICM. Intensity of fluorescence was analysed in each cell. Scale bar 20 µm. Statistical analysis was performed using one-way ANOVA, $p > 0.05$.

that RUFY4 might also interact with these organelles, involved like ER, in supplying membranes for autophagosome biogenesis [25].

## 2.3. RUFY4 co-localizes with mitochondria upon macrophage activation

Immunofluorescence microscopy indicates that RUFY4 localizes to distinct and abundant subcellular organelles in activated macrophages. We, therefore, tested if these organelles could correspond to mitochondria as inferred from our MS analysis. Antibodies raised against mitochondrial intermembrane-associated apoptosis-inducing factor (AIF) and succinate dehydrogenase complex flavoprotein subunit A (SDHA) were used to visualize mitochondria by microscopy in RAW macrophages, AM and IL4-bmDC stimulated or not with transfected pcDNA or LPS (figure 4*a*). At the phenotypical level, the mitochondrial network of the different cells did not look affected by the

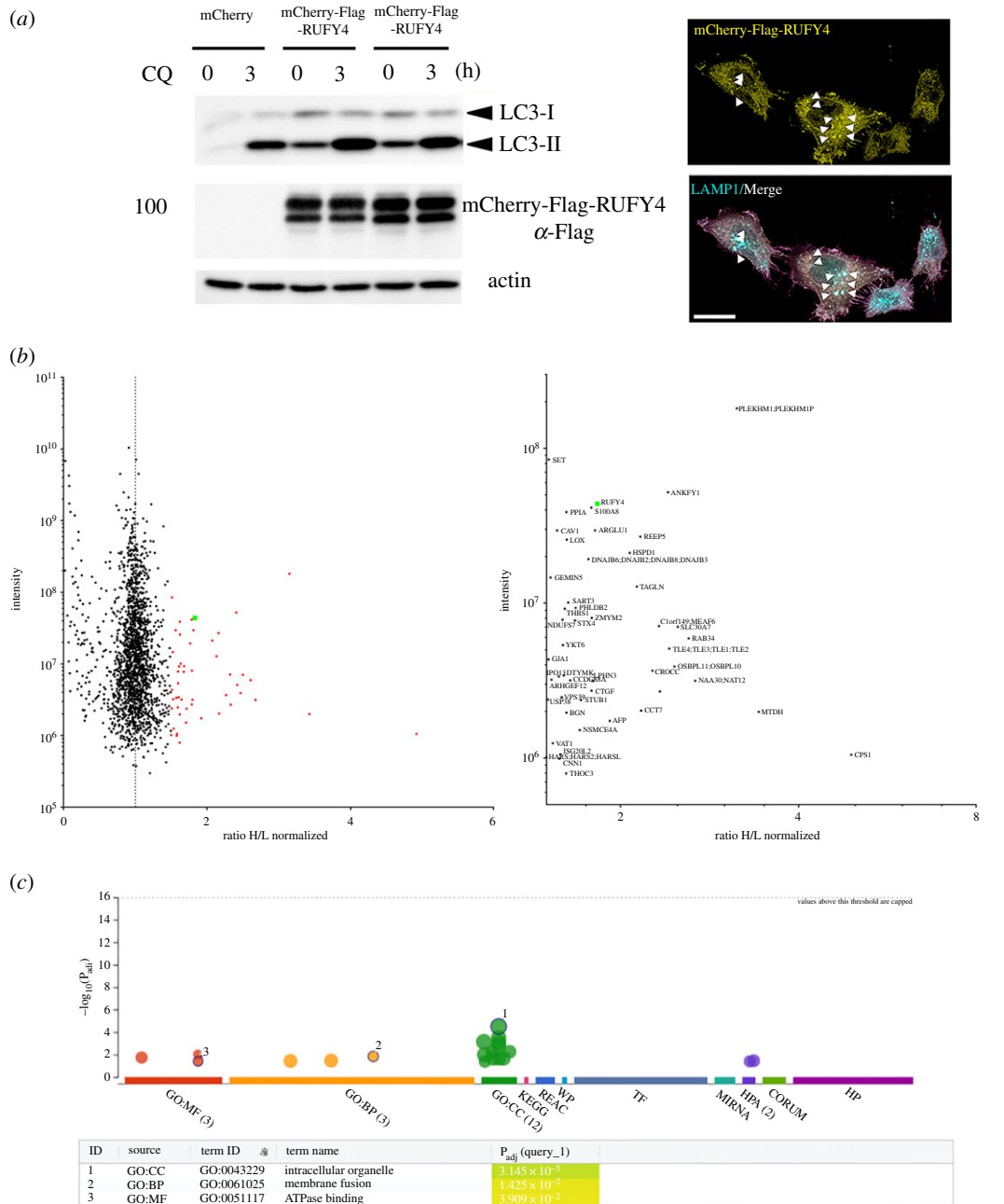

**Figure 3.** Identification of RUFY4 interacting molecules. (*a*) HeLa cells were stably transfected with a mCherry-flag-tagged RUFY4 construct. Autophagy flux in RUFY4 expressing cells was monitored by immunoblotting of LC3b I and LC3b II accumulation at steady state or upon 3 h of chloroquine (CQ) treatment. Clustering of mCherry-RUFY4 (yellow) and LAMP1 (cyan)-positive organelles and co-localization (white, arrowheads) by ICM is shown on the right. Scale bar 20 µm. (*b*) SILAC-based mass spectrometric analysis of RUFY4 interactome after immunoprecipitation. A normalized heavy/light (H/L) ratio ≥ 1.5 was used as a cut-off to identify potential RUFY4-mCherry interaction partners (right and electronic supplementary material, table S1). (*c*) Gene ontology analysis using the G-Profiler program of putative RUFY4 interacting partners indicating association with intracellular organelles including endosomes and mitochondria (electronic supplementary material, table S1). Data are available via ProteomeXchange with identifier PXD026728.

activation process, although it has been shown to trigger an energy metabolism switch from respiration to glycolysis in the time frame studied [26]. As expected, only activated cells displayed RUFY4 staining, which co-localized with the mitochondrial network of the different cell types examined (figure 4*a*). LAMP1- and LAMP2-positive late endosomal and lysosomal compartments used as a co-localization control were found not to be associated with RUFY4 irrespective of the activation state of the cells (figure 4*b*) and contrasting with the situation observed upon ectopic expression of the protein (figure 3*a*).

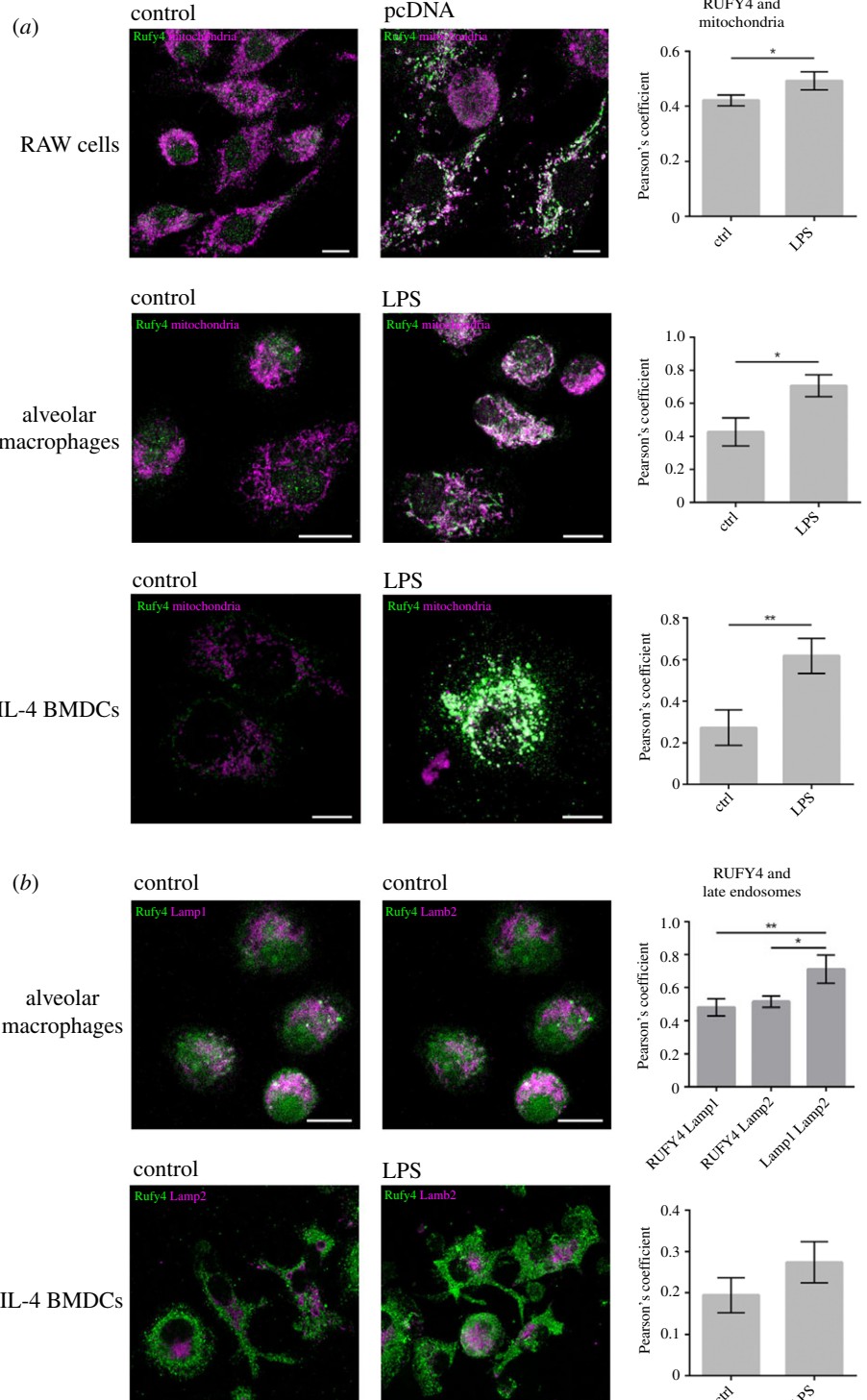

**Figure 4.** RUFY4 co-distributes with mitochondria in activated phagocytes. (a) RAW cells, alveolar macrophages (AM) and bone marrow-derived dendritic cells (BMDC) grown in IL-4 were activated by transfection of empty pcDNA3.1 plasmid (RAW) or by 100 ng ml$^{-1}$ of LPS for 8 h (AM, BMDC) and stained for RUFY4 and mitochondrial markers AIF (RAW and AM) and SDHA (BMDC) and imaged by ICM. Co-localization of RUFY4 with mitochondria was quantified using Pearson's coefficient correlation in IMAGEJ and statistical relevance established using Student's t-test, $^*p \leq 0.05$, $^{**}p \leq 0.01$. Scale bars are 10 μm. (b) AM were stained for RUFY4, LAMP1 and LAMP2 and imaged by ICM. BMDC were activated by 100 ng ml$^{-1}$ of LPS for 6 h and stained for RUFY4 and LAMP2. As expected, only LAMP1 and LAMP 2 showed a significant degree of co-localization together, but not with RUFY4. Co-localization was quantified using Pearson's coefficient correlation in IMAGEJ and statistical relevance established using one-way ANOVA, $^*p \leq 0.05$, $^{**}p \leq 0.01$. Scale bar 10 μm.

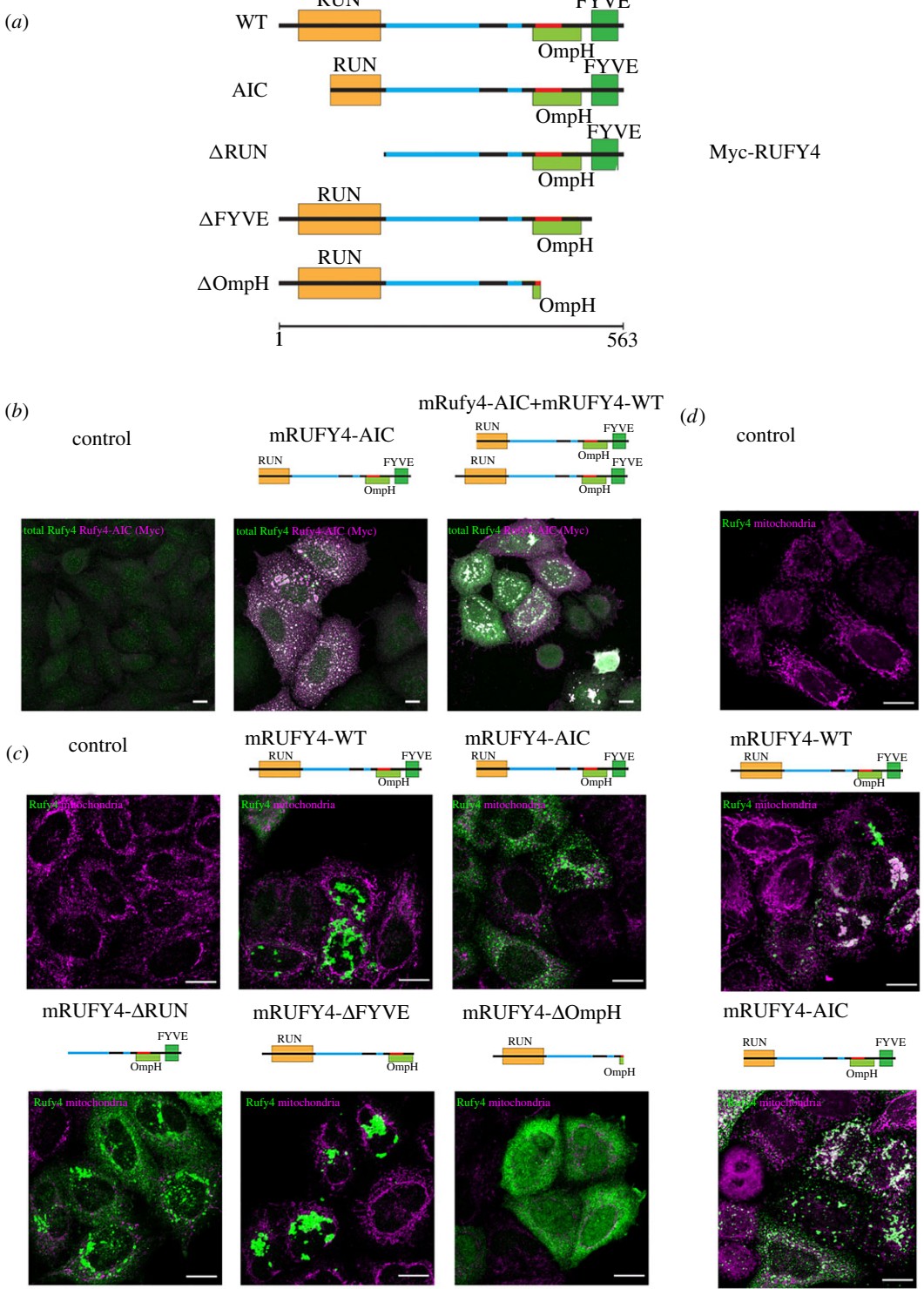

**Figure 5.** Ectopic expression of RUFY4 mutants induces mitochondria clustering with different efficacy. (*a*) Schematic of RUFY4 mutants constructs. (*b*) HeLa cells were transfected with Myc-RUFY4 WT, Myc-RUFY4 AIC or the two together and stained for RUFY4 and Myc prior to ICM. Scale bar 10 μm. (*c*) HeLa cells were transfected with indicated RUFY4 mutants and stained for RUFY4 and the mitochondrial marker (AIF) prior to ICM. Scale bar 20 μm. (*d*) HeLa cells were transfected with indicated RUFY4 mutants and stained for RUFY4 and Mitotracker Deep Red prior to ICM. Scale bar 20 μm.

## 2.4. The subcellular distribution of truncated forms of RUFY4 is affected

RUFY4 bears a FYVE domain in its C-terminal portion, that mediate interaction with PtdIns(3)*P*, separated by two CC domains from its N-terminal RUN domain (figures 1*c* and 5*a*). In previous

studies [4], we have shown that ectopically expressed RUFY4 in HeLa cells is primarily localized in the vicinity of enlarged autophagosomes and tethered lysosomes (figure 3a). The RUN domain of RUFY4 is absolutely required for this process, while its FYVE domain was only required for autophagosome enlargement, but not lysosome tethering. During our investigations on the structural organization of RUFY4, we identified a putative Skp/OmpH domain immediately upstream of the C-terminal FYVE zinc finger (electronic supplementary material, figure S1B). The 17 kDa protein (Skp/OmpH) of *Escherichia coli* is a homotrimeric periplasmic chaperone for newly synthesized outer-membrane proteins, which also interacts directly with bacterial membrane lipids and LPS [27]. The homology domain is known to fold into short α-helices that partially constitutes the limited hydrophobic core of Skp/OmpH that mediates its oligomerization into trimers [27]. A new truncated form of RUFY4 extending the deletion of its C-terminus to this putative OmpH domain was engineered (ΔFYVE full) (figure 5a). Moreover, we also designed a construct forcing the translation of RUFY4 from the AIC identified in exon 4 of the *rufy4* gene and thus producing a protein amputated from about one-third of its N-terminal RUN domain (figure 5a). All wild-type (WT) and myc-tagged mutant forms of RUFY4 were expressed ectopically in HeLa cells (figure 5b,c) and their distribution established by ICM.

We first tested if WT RUFY4 and AIC could associate and potentially co-distribute when expressed co-temporally. Detection of the two differently tagged forms of RUFY4 showed perfect co-localization confirming that the two isoforms are likely to associate, potentially dimerize and modulate the activity of RUFY4 (figure 5b). We next monitored the co-distribution of the different RUFY4 mutants with mitochondria in HeLa cells. Little if any co-localization could be observed in a majority of cells (figure 5c), although the localization of the different mutants recapitulated our previous observations, with WT RUFY4 causing organelle clustering and tethering (figure 5c). RUN domain-deleted (ΔRUN) and the AIC constructs had a similar pattern of distribution, causing less acute clustering and suggesting that what remains of the RUN domain in the RUFY4 AIC is inactive. Importantly, deletion of the FYVE domain creates a mutant (ΔFYVE), that upon overexpression, causes a detectable aggregation of organelles including the ER and mitochondria [5]. This phenotype was recapitulated upon expression of the ΔFYVE mutant (figure 5b), but extension of the deletion to the C-terminal OmpH domain (ΔOmpH) resulted in a diffuse distribution of the protein and abolished completely organelle collapse. These observations suggest that the region containing the putative OmpH domain promotes unregulated organelle binding upon deletion of the FYVE domain present in RUFY4. The OmpH domain could directly facilitate mitochondrial membrane binding of the protein in absence of regulated PtdIns(3)*P* binding capacity or potentially contribute to RUFY4 dimerization, which is probably required for its function as inferred from our structural model and examples from other FYVE-bearing molecules [28]. In HeLa cells, ectopic RUFY4 seems, therefore, preferentially to be associated with autophagosome, late endosomes and ER in agreement with the identification by MS of PLEKHM1, RAB34 and YKT6 as potential RUFY4 interacting partners. However, occasionally, we could observe the strong association of over-expressed WT RUFY4 and AIC with mitochondria (figure 5d). RUFY4 and its alternate AIC form seem, therefore, to behave heterogeneously upon ectopic expression, further suggesting that RUFY4 function might be dependent on a tight dosage or on cell-specific molecular partners or post-translational modifications that regulate its binding to mitochondria upon cell activation by MAMPs.

## 2.5. Analysis of alveolar macrophages deleted in *rufy4* exon 3 reveals the functionality of the alternative translation initiation codon

To further explore the physiological relevance of the alternatively translated truncated form of RUFY4, we generated a novel transgenic mouse model with floxed alleles for *rufy4*. This modification at the borders of the exon 3 of the *rufy4* gene, allows, upon Cre recombinase expression, the deletion of this exon that prevents the expression of the full-length protein, but still allow the translation of the mRNA from the AIC in exon 4 (figure 6a). *rufy4Δex3*[lox/lox] C57/BL6 mice were crossed with a Itgax-cre deleter strain [29] to specifically inactivate *rufy4*[lox/lox] in CD11c-expressing cells, including DC subsets and AM. We could confirm the deletion of WT *rufy4* and the expression of *rufy4Δex3* mRNA by qPCR in both AM and bone-marrow-derived DC (figure 6b). We then submitted WT and *rufy4* AIC AM or DCs to ICM and confirmed the translation of RUFY4 AIC in physiological conditions (figure 6c). Like for full-length RUFY4, RUFY4 AIC co-localization to mitochondria was increased by LPS activation in AM. Importantly, when mitochondria association was quantified in DCs (figure 7a), RUFY4 AIC was already co-localized with mitochondria in non-activated *rufy4Δex3* cells to a level similar to those reached upon LPS activation of control DCs. These observations indicate that partial RUN domain deletion in RUFY4 AIC could enhance

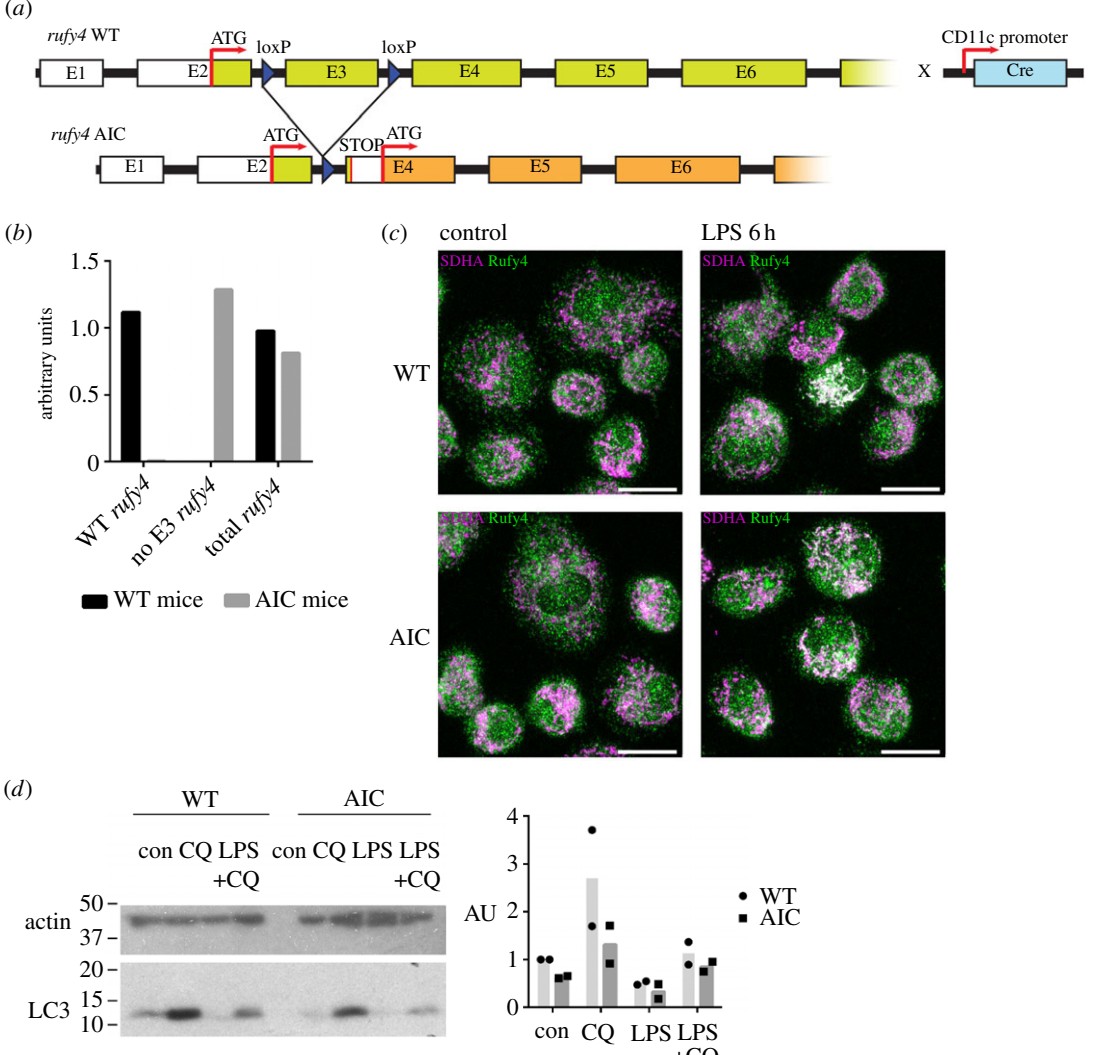

**Figure 6.** *rufy4* AIC is physiologically expressed in alveolar macrophages. (*a*) Schematic of the gene re-organization in the transgenic mice expressing only *rufy4* AIC. The third exon of the *rufy4* gene was flanked by two loxP sequences to promote its excision upon Cre recombinase expression and introduce a stop after exon 2. *rufy4*<sup>lox/lox</sup> mice were bred to Itgax-Cre+ mice [29] to express *rufy4* AIC in CD11C-expressing DC and AM (*rufy4* AIC mouse). (*b*) Amount of *rufy4* mRNA in WT and *rufy4* AIC AM was monitored by qPCR using primers detecting the different transcripts of *rufy4*. ND, not detected. (*c*) Alveolar macrophages from WT and *rufy4* AIC mouse were activated by 100 ng ml⁻¹ of LPS for 8 h and stained for RUFY4 (green) and mitochondrial SDHA (lilac) and imaged for co-localization (white) by ICM. Scale bar 10 µm. (*d*) Autophagy flux in WT and *rufy4* AIC alveolar macrophages was monitored by immunoblotting of LC3b I and LC3b II accumulation at steady state or upon 4 h of chloroquine (CQ) treatment. AM activation was performed with 100 ng ml⁻¹ of LPS for 8 h prior autophagy monitoring. Actin is used as loading control. Quantification by densitometry of ratio of LC3b to actin is shown on the right ($n = 2$).

association with the mitochondria network, further suggesting that LPS activation could functionally inhibit the RUFY4 RUN domain activity to promote the association to mitochondria in AM and DC. We next monitored the levels of autophagy flux in both *rufy4Δex3* (AIC) AM (figure 6*d*) and DCs (figure 7*b*). LC3b I processing and LC3b II accumulation measured upon chloroquine or bafilomycin treatment was found equivalent in AIC and WT cells. RUN domain deletion in RUFY4 does, therefore, seem not to impact autophagy in the studied cells, in line with the lack of observed interaction between RUFY4 and endosomes (figure 4*b*). Given the co-localization of RUFY4 AIC and mitochondria, we next monitored the mitochondrial status using mitochondria-specific fluorescent labels that distinguish respiring (Mitotracker Deep Red) from damaged mitochondria by flow cytometry [30]. Cytometry analysis suggested that *rufy4Δex3* (AIC) DCs display less damaged mitochondria than WT cells at steady state and more respiratory ones upon LPS stimulation. Although large experimental variations decreased the statistical relevance to this trend, these results suggest that expression of RUFY4 AIC alone in DCs might increase damaged mitochondrial clearance.

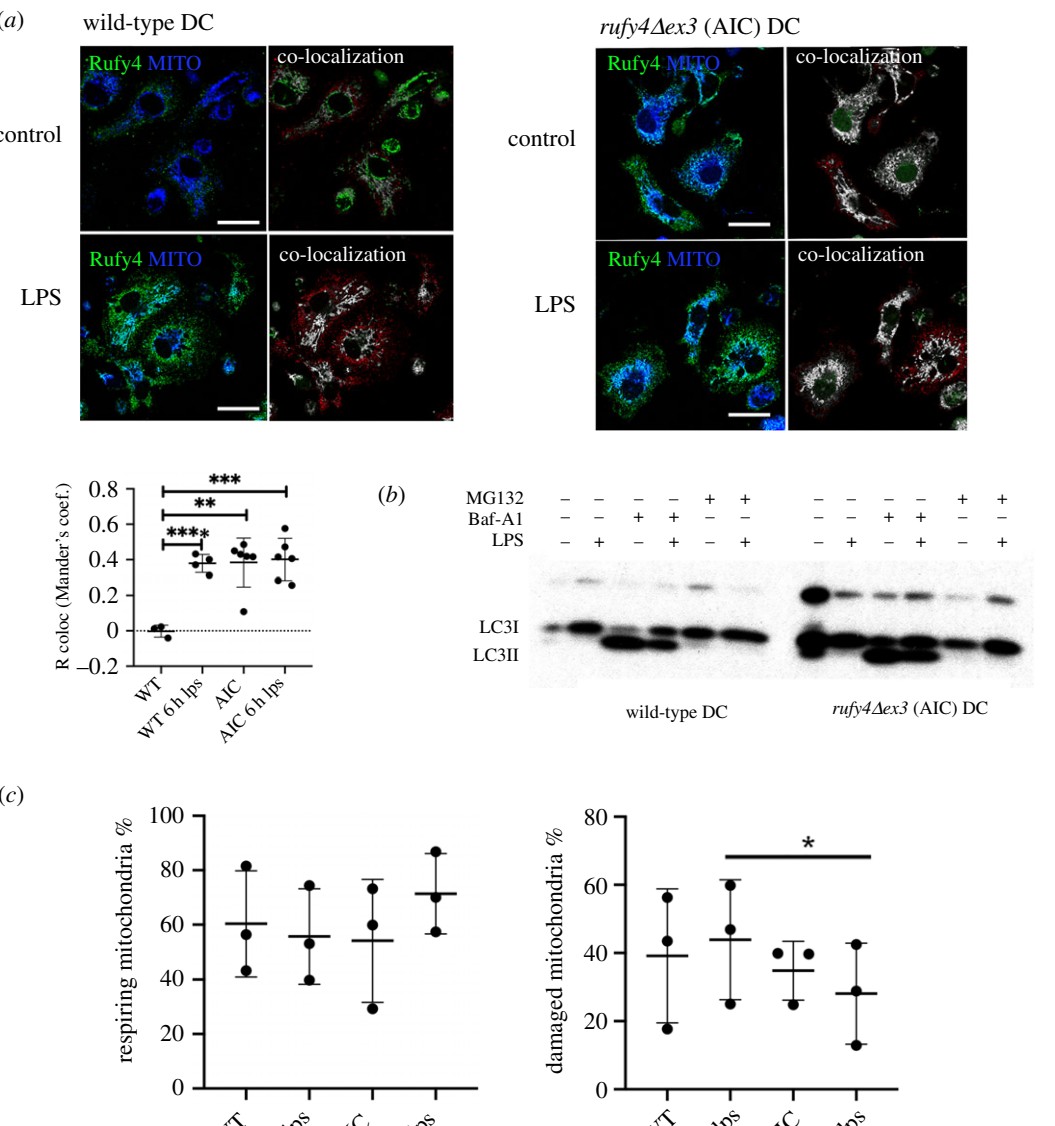

**Figure 7.** *rufy4* AIC is associated with mitochondria in bone marrow-derived dendritic cells (BMDCs). (*a*) BMDCs from WT and *rufy4* AIC mice were activated by 100 ng ml⁻¹ of LPS for 6 h and stained for RUFY4 (green) and mitochondrial SDHA (blue) and imaged for co-localization (white) by ICM using voxel gating with the 'Coloc' tool from the IMARIS. Scale bar 20 µm. R correlation was performed using Mander's coefficient and Student's *t*-test, $^{**}p \leq 0.01$, $^{***}p \leq 0.001$, $^{****}p \leq 0.0001$. (*b*) Autophagy flux in WT and *rufy4* AIC BMDC was monitored by immunoblotting of LC3b I and LC3b II accumulation at steady state or upon 4 h of 100 nM bafilomycine A1 (Baf-A1)) treatment. DC activation was performed with 100 ng ml⁻¹ of LPS for 6 h prior autophagy monitoring. Proteasome inhibition with 5 µM MG132 was used as control. (*c*) Quantitative analysis of damaged and respiring mitochondria by flow cytometry using MitoTracker staining of steady state and LPS activated WT and *rufy4* AIC mouse BMDCs. Statistical relevance established using Student's *t*-test, $^{*}p \leq 0.05$.

## 3. Discussion

MAMPs triggering of TLR4, as well as interferons or IL-1ß exposure promotes phagocytes activation resulting in secretion of inflammatory cytokines and enhanced antigen processing and presentation. The signal transduction pathways mediating this activation processes are complex and coordinate novel gene transcription events with intense changes in protein synthesis, membrane trafficking, actin organization and energetic metabolism [31]. It is, therefore, expected that molecules like RUFY4 that have a pattern of expression restricted to immune cells and play a regulatory role in different aspects of membrane traffic, would be functionally regulated upon DC or macrophages activation by MAMPs like LPS.

The *rufy4* gene is located on chromosome 1 in mouse (Chr 2 in human), and its promotor region is transcriptionally active upon GM-CSF and IL-4 exposure explaining its strong expression in AM and monocyte-derived DC. *rufy4* mRNA expression was found to be decreased upon MAMPs detection, however, enhanced association with mitochondria was observed in these conditions. This association was confirmed by MS identification of several mitochondrial proteins among the interacting partners of ectopically expressed RUFY4. In HeLa cells, mitochondria are, however, not the main organelles targeted by over-expressed RUFY4, that rather promotes membrane organelle tethering, as further inferred by the identification of VPS39, RAB34, YKT6 and PLEKHM1 as RUFY4 interacting partners. In phagocytes, however, the consequences of endogenous RUFY4 association with mitochondria remain unclear, but the role of these organelles in energy production, as well as a source of membranes for autophagy [25], could indicate that RUFY4 plays a regulatory role of mitochondria interactions with others membrane organelles or damaged mitochondria clearance upon LPS activation of these cells.

We have shown that RUFY4 exists as two translationally regulated isoforms, owing to the existence of an AIC in exon 4 that allows the synthesis of a truncated RUFY4 lacking a functional RUN domain in its N-terminal part. Interestingly in humans, one of RUFY4 transcripts lacking entirely the RUN domain was also identified in the databases. These two alternatively translated isoforms are likely to be expressed contemporarily and have the ability to interact with each other. Whether they dimerize together through the different CC or putative OmpH domains present in their middle segments remains, however, to be evaluated. Importantly, we have demonstrated that the AIC form of RUFY4 can be expressed physiologically in AM or bone-marrow-derived DC. The absence of a fully functional RUN domain in the AIC form seems to enhance RUFY4 association to mitochondria. The observation that, in non-activated *rufy4Δex3* DCs, RUFY4 is localized to the mitochondria irrespective of the activation state of DCs, further supports that the regulated expression of this isoform could influence RUFY4 targeting to mitochondria upon MAMPs stimulation of AM or DCs.

RUFY4 expression has been shown to increase the resistance of cells to intracellular bacterial infection such as *B. abortus* [4] and of *S. typhimurium* [7]. The reason of this inhibition remains to be established, however, the rapid kinetics observed for bacterial clearance suggest that exacerbated xenophagy by RUFY4 could be key for this process. Given the similarities between xenophagy and mitophagy [32], RUFY4 could have an important role in both of these functions in AM and DC. We showed that the OmpH domain in the C-terminal part of RUFY4 impacts the capacity of RUFY4 to bind mitochondria in the absence of a functional FYVE region. Skp/OmpH's role as periplasmic chaperone that assists bacterial outer-membrane proteins in their folding and insertion into membranes, suggests that RUFY4 could be capable of binding mitochondrial membrane in the context of PtdIns(3)P enrichment and could play an active role during mitophagy. Indeed, it has been proposed that damaged mitochondria are ubiquitinated and dynamically encased in ER layers, providing sites for mitophagosomes formation [33]. These sites are likely to be the same specialized ER domains responsible for *S. typhimurium* autophagy via PtdIns(3)P accumulation [34]. The regulation of RUFY4 expression and function is, therefore, far more complex than anticipated. Its translational regulation and dependence on LPS activation for associating with mitochondria in physiologically relevant cells has considerably complexified the dissection of its molecular function. Our observations point nevertheless to an adaptor function interfacing PtdIns(3)P-enriched domains with the membrane fusion and tethering machinery leading to selective mitochondria targeting.

# 4. Methods

## 4.1. Mice

WT female C57BL/6 mice were purchased from Janvier, France. *Rufy4Δex3^{loxp/loxp}* mice were developed at the Centre d'Immunophénomique (CIPHE, Marseille, France). *Rufy4Δex3^{loxp/loxp}* were crossed with Itgax-Cre+ mice [29] and backcrossed, to obtain stable homozygotic lines for the loxp sites expressing Cre.

## 4.2. Cell culture

Bone-marrow-derived DC were cultured with GM-CSF as described previously [35]. AM and RAW cells were cultured as described previously [36]. Ten week-old mice were euthanized, their rib cage removed and small incision was cut in the upper part of trachea. Lungs were washed at least 10 times by 1 ml of

phosphate buffered saline (PBS) + 2% fetal calf serum (FCS) + 2 mM EDTA. The washing medium containing cells was kept on ice in 50 ml falcon tubes containing 10 ml of AM medium (Roswell Park Memorial Institute medium (RPMI) + 10% FCS + 1% pen/strep + 1% pyruvate + 1% glutamine). Cells were then centrifuged at 1500 rpm, 4°C for 5 min and red blood cells lysis was performed on ice using red blood cells lysis buffer (eBioscience, 00-4333-57) according to the manufacturer's instructions. Cells were then resuspended in AM medium and seeded in uncoated 6-well plates (Thermo Scientific, 150239). Cells were grown in AM medium with media supplemented with 2.5% GM-CSF. HeLa cells were maintained in Dulbecco's modified eagle medium (Gibco Invitrogen) supplemented with 10% FCS (Hyclone, PERBIO), at 37°C and 5% $CO_2$.

## 4.3. Immunodetection and immunoprecipitation

A 25–50 µg of TX-100 soluble material was separated by 3–15% gradient or 12% SDS–PAGE prior immunoblotting and chemiluminescence detection (Pierce). Antibodies used in this study were anti-RUY4 raised in rabbit against peptides 375–391 and 454–470 of mouse RUFY4. Mouse Anti-Myc (9B11, Cell Signaling), mouse Anti-Flag (M2, Sigma), rat anti-LAMP1 (134B, Biolegend), mouse anti-AIF (E-1, Santa Cruz), mouse anti-SDHA (2E3GC12, Abcam), mouse anti-LC3 (2G6, NanoTools), mouse anti-ß-actin (AC-15, Sigma). Secondary antibodies were from Jackson Immunoresearch, Molecular Probes (USA) and from Cell Signaling Technology. For immunofluorescence, cells on coverslips were fixed with 3.5% paraformaldehyde and permeabilized with 0.1% Triton X-100. Images were taken by a Zeiss LSM780 or Leica SP5 confocal microscope using 63× or 40× objective. Processing and quantification was performed using FIJI software [37]. Co-localization was quantified using JACoP plugin [38]. Statistical analysis was performed using GRAPHPAD PRISM. For two sets of values, we used *t*-tests, for multiple sets of values one-way ANOVA. $^*p \leq 0.05$, $^{**}p \leq 0.01$, $^{***}p \leq 0.001$, $^{****}p \leq 0.0001$. MitoTracker DeepRed and MitoTracker Green staining was performed according to the manufacturer's instructions (Thermofhisher) and detected by flow cytometry.

## 4.4. Complementary DNA cloning, *in vitro* transcription and gene transduction

Mouse *rufy4* cDNA was amplified by PCR using complementary DNA (cDNA) from DC as template, and then cloned to a pcDNA3.1 vector (invitrogen) with tagging with myc (N-terminus), FLAG (C-terminus) or mCherry (N-terminus). Previously prepared plasmids or cDNA from AM or bone marrow-derived dendritic cells were used as PCR templates. Q5 hot start polymerase was used for PCR. Cloning was performed using InFusion kit (Takara) according to the manufacturer's instructions. Plasmids containing truncated forms of *rufy4* were created by amplifying by PCR the whole WT *rufy4* plasmid except the part of the gene to be removed. Primers contained 7 and 8 bp long sequences from the end of other plasmids, creating a 15 bp long homologous sequence on the ends of the PCR product as a substrate for the InFusion HD Enzyme (Takara). The plasmid DNAs were introduced to the cell lines with the use of JetPrime reagent (Polyplus).

## 4.5. Quantitative polymerase chain reaction

Total mRNA was purified using the RNeasy Mini Kit (Qiagen); 100 ng to 1 µg of total RNA were subjected to reverse transcription using SuperScript II. Each gene transcripts were quantified by SYBR Green method with 7500Fast (Applied Biosystems). The relative amount of each transcript was determined by normalizing to internal housekeeping gene expression (gapdh). See a list of primers in the electronic supplementary material, table S2.

## 4.6. Mass spectrometry

For SILAC labelling, mCherry-RUFY4 expressing HeLa cells were cultured in media supplemented with either L-arginine-12C614N4 (Arg0) and L-lysine-12C614N2 (Lys0) or L-arginine-13C615N4 (Arg10) and L-lysine-U-13C615N2 (Lys8) as described previously [19]. SILAC labelled cells were lysed using GTPase lysis buffer. Cells were cultured in heavy SILAC labelled media, Cherry-tag immunoprecipitated, eluates were mixed 1 : 1 (v/v) and run on SDS–PAGE. The gel lane was cut into 10 slices which were in-gel digested by trypsin and liquid chromatography-tandem MS analyses were performed on an EasyLC nano-HPLC coupled to an Orbitrap Elite mass spectrometer (bothThermo Scientific). The MS data of all SILAC experiments were processed using default parameters of the MAXQUANT software (1.3.0.5) [39].

The MS proteomics data have been deposited to the ProteomeXchange Consortium via the PRIDE [1] partner repository with the dataset identifier PXD026728.

## 4.7. Mitochondrial staining

Determination of respiratory chain damage was performed by double staining with two different mitochondria-specific dyes, MitoTracker Green FM (516 nm) and MitoTracker Deep Red FM (665 nm) (Thermo Fisher Scientific, ref. M7514 and M22426), to distinguish total and respiring mitochondria, respectively. MitoTracker Deep Red FM enters an actively respiring cell, where it is oxidized as the corresponding red fluorescence probe and sequesters in the mitochondria. The treated cells were incubated with 100 nM MitoTracker Green FM and 100 nM MitoTracker Deep Red FM (diluted in warm RPMI) in the dark at 37°C for 15 min at the end of the treatment period. Cells were harvested and pellets were resuspended in 0.5 ml of PBS, prior immediate analysis by flow cytometry.

Ethics. For all studies, age-matched WT and transgenic 6–10 weeks female mice were used. All animals were maintained in the animal facility of CIML or CIPHE under specific pathogen-free conditions accredited by the French Ministry of Agriculture to perform experiments on live mice. These studies were carried out in strict accordance with the Guide for the Care and Use of Laboratory Animals of the European Union. All experiments were approved by the Comité d'Ethique PACA and MESRI (approval number APAFIS#18981-2019020710111763). All efforts were made to minimize animal suffering.

Data accessibility. The MS proteomics data have been deposited to the ProteomeXchange Consortium via the PRIDE partner repository with the dataset identifier PXD026728.

Authors' contributions. J.V. and P.P. contributed equally to the design and implementation of the research and writing of the manuscript. J.V., V.C., S.T., Z.L., E.S. performed research. D.G.M.E. and I.D. performed the mass spectrometry analysis. E.G., C.R.A., B.S. and Y.L. contributed to the experimental design.

Competing interests. The authors declare to have no competing interest.

Funding. The PP laboratory is 'Equipe de la Fondation de la Recherche Médicale' (FRM) sponsored by the grant DEQ20180339212. The laboratory received financial support from the Innate Immunocytes in Health and Disease (I2HD) collaborative project between CIML, AVIESAN and SANOFI. The project and J.V. were supported by grants from l'Agence Nationale de la Recherche (ANR) 'DCBIOL Labex ANR-11-LABEX-0043', 'INFORM Labex ANR-11-LABEX-0054' funded by the 'Investissements d'Avenir' French government programme. This work was also financially supported by the project PTDC/BIA-CEL/28791/2017 and POCI-01-0145-FEDER-028791, as well as POCI-01-0145-FEDER-030882 and PTDC/BIA-MOL/30882/2017, funded by FEDER, through COMPETE2020—Programa Operacional Competitividade e Internacionalização (POCI), and by national funds (OE), through FCT/MCTES.

Acknowledgements. We thank the Maratona da Saúde for its support. We acknowledge the support of the CIML imaging core facility 'ImagImm' supported by ANR-10-INBS-04-01 France Bio Imaging for advanced microscopy.

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
