## [Peer Review File · Royal Society Open Science]

Review History

RSOS-202333.R0 (Original submission)

Review form: Reviewer 1

Is the manuscript scientifically sound in its present form?

Yes

Are the interpretations and conclusions justified by the results?

Yes

Is the language acceptable?

Yes

Do you have any ethical concerns with this paper?

No

Have you any concerns about statistical analyses in this paper?

No

Recommendation?

Accept as is

Comments to the Author(s)

Valecka et al report on the characterization in myeloid cells of the FYVE containing protein Rufy4 and its regulation. They found that the protein is well expressed in alveolar macrophages and MDDC upon activation by LPS. In addition, they characterize by SILAC the Rufy4 interacting proteins. Among them PLEKHM1 and rab34, suggesting a role in mitochondria and late endosome regulation. Interestingly, they observe that there are two isoforms of the protein expressed due to the use of alternative codon initiation for translation. The shorter one lacks the N-ter part starts immediately in the RUN domain. Moreover, by performing a structure-function analysis, they characterize a new domain in the protein involved in effect on organelles aggregation. They nicely show further that upon LPS stimulation Rufy4 is recruited to the mitochondria.

The report is well written, and the data are very convincing. They bring important information regarding the protein Rufy4 and its role in myeloid cells upon activation. The paper deserves to be published as it is.

Minor comment. Fig 1D: individual points should be displayed in addition to the mean and SD.

Review form: Reviewer 2

Is the manuscript scientifically sound in its present form?

Yes

Are the interpretations and conclusions justified by the results?

Yes

Is the language acceptable?

Yes

Do you have any ethical concerns with this paper?

No

Have you any concerns about statistical analyses in this paper?

No

Recommendation?

Accept with minor revision (please list in comments)

Comments to the Author(s)

In this manuscript, Valecka et al show that the expression of Rufy4, a protein of the RUN and FYVE family is regulated by alternative splicing mechanisms. This protein is also shown to associate to mitochondria in response to stress.

This study provides interesting new knowledge about the regulation, localization and potential function of Rufy4. This is of particular interest considering that members of the RUN and FIVE

protein family have been shown to play a role in autophagy. The regulated interaction with endocytic organelles and mitochondria is of particular interest.

The manuscript is generally well-written and presents interesting new aspects of the Rufy4 biology. Interesting new data with proteomics and the identification of binding protein partners are also provided.

Minor comments:

-At the end of page 3, "antigen phagocytosis" is not appropriate as antigens are not directly phagocytosed but rather generated through processing in the phagosome lumen.

-In the introduction, on page 4, in the sentence that starts with "A role for Rufy4 in late endosomal organelles", it should be indicated that these proteins are known to be associated with late endosomes.

-In the Results section, at the end of the first paragraph, "by qPCR and"...and what?

-In the Results section, on page 5, the sentence "We were however..." is difficult to read and should be rewritten (with "however, between commas).

-In the Results section, on page 8, it would be best to change "...cause the massive aggregation of organelles including of ER and mitochondria" to "...cause a detectable aggregation of organelles including the ER and mitochondria".

Decision letter (RSOS-202333.R0)

Dear Dr Pierre

On behalf of the Editors, we are pleased to inform you that your Manuscript RSOS-202333 "Rufy4 exists as two translationally regulated isoforms, that localize to the mitochondrion in activated macrophages." has been accepted for publication in Royal Society Open Science subject to minor revision in accordance with the referees' reports. Please find the referees' comments along with any feedback from the Editors below my signature.

Please submit your revised manuscript and required files (see below) no later than 7 days from today's (ie 14-May-2021) date. Note: the ScholarOne system will 'lock' if submission of the revision is attempted 7 or more days after the deadline. If you do not think you will be able to meet this deadline please contact the editorial office immediately.

Please note article processing charges apply to papers accepted for publication in Royal Society Open Science (<https://royalsocietypublishing.org/rsos/charges>). Charges will also apply to

papers transferred to the journal from other Royal Society Publishing journals, as well as papers submitted as part of our collaboration with the Royal Society of Chemistry (<https://royalsocietypublishing.org/rsos/chemistry>). Fee waivers are available but must be requested when you submit your revision (<https://royalsocietypublishing.org/rsos/waivers>).

on behalf of Dr Simon Cook (Associate Editor) and Malcolm White (Subject Editor)
openscience@royalsociety.org

Reviewer comments to Author:
Reviewer: 1

Comments to the Author(s)

Valecka et al report on the characterization in myeloid cells of the FYVE containing protein Rufy4 and its regulation. They found that the protein is well expressed in alveolar macrophages and MDDC upon activation by LPS. In addition, they characterize by SILAC the Rufy4 interacting proteins. Among them PLEKHM1 and rab34, suggesting a role in mitochondria and late endosome regulation. Interestingly, they observe that there are two isoforms of the protein expressed due to the use of alternative codon initiation for translation. The shorter one lacks the N-ter part starts immediately in the RUN domain. Moreover, by performing a structure-function analysis, they characterize a new domain in the protein involved in effect on organelles aggregation. They nicely show further that upon LPS stimulation Rufy4 is recruited to the mitochondria.

The report is well written, and the data are very convincing. They bring important information regarding the protein Rufy4 and its role in myeloid cells upon activation. The paper deserves to be published as it is.

Minor comment. Fig 1D: individual points should be displayed in addition to the mean and SD.

Reviewer: 2

Comments to the Author(s)

In this manuscript, Valecka et al show that the expression of Rufy4, a protein of the RUN and FYVE family is regulated by alternative splicing mechanisms. This protein is also shown to associate to mitochondria in response to stress.

This study provides interesting new knowledge about the regulation, localization and potential function of Rufy4. This is of particular interest considering that members of the RUN and FIVE protein family have been shown to play a role in autophagy. The regulated interaction with endocytic organelles and mitochondria is of particular interest.

The manuscript is generally well-written and presents interesting new aspects of the Rufy4 biology. Interesting new data with proteomics and the identification of binding protein partners are also provided.

Minor comments:

-At the end of page 3, “antigen phagocytosis” is not appropriate as antigens are not directly phagocytosed but rather generated through processing in the phagosome lumen.

-In the introduction, on page 4, in the sentence that starts with “A role for Rufy4 in late endosomal organelles”, it should be indicated that these proteins are known to be associated with late endosomes.

-In the Results section, at the end of the first paragraph, “by qPCR and”...and what?

-In the Results section, on page 5, the sentence “We were however...” is difficult to read and should be rewritten (with “however, between commas).

-In the Results section, on page 8, it would be best to change “...cause the massive aggregation of organelles including of ER and mitochondria” to “...cause a detectable aggregation of organelles including the ER and mitochondria”.

===PREPARING YOUR MANUSCRIPT===

===PREPARING YOUR REVISION IN SCHOLARONE===

Author's Response to Decision Letter for (RSOS-202333.R0)

See Appendix A.

Decision letter (RSOS-202333.R1)

Dear Dr Pierre,

I am pleased to inform you that your manuscript entitled "Rufy4 exists as two translationally regulated isoforms, that localize to the mitochondrion in activated macrophages." is now accepted for publication in Royal Society Open Science.

on behalf of Dr Simon Cook (Associate Editor) and Malcolm White (Subject Editor)

Marseille, 7 juillet 2021

Dear Monitoring editor,

Please find a revised version of our manuscript entitled “Rufy4 exists as two translationally regulated isoforms, that localize to the mitochondrion in activated macrophages” by Valecka et al., which was accepted for publication in Royal Society Open Science. We thank the reviewers for their extremely positive comments on our work and have addressed all the minor corrections that were suggested in their reviews, including modifications of Figure 1D. We have now included the Pride Proteomics Database entry number requested to proceed with the paper publication.

With my best regards,

Dr. Philippe PIERRE

Head of the Dendritic Cell Biology Laboratory, Centre d'Immunologie de Marseille-Luminy

Directeur de recherche CE, CNRS,